# Antibacterial Discovery: 21st Century Challenges

**DOI:** 10.3390/antibiotics9050213

**Published:** 2020-04-28

**Authors:** Paul S. Hoffman

**Affiliations:** Division of Infectious Diseases and International Health, Department of Medicine, University of Virginia, Charlottesville, VA 22908, USA; psh2n@virginia.edu; Tel.: +1-434-806-7750

**Keywords:** antibiotic, antibiotic resistance, drug targets, amixicile, pathogen group specific targets

## Abstract

It has been nearly 50 years since the golden age of antibiotic discovery (1945–1975) ended; yet, we still struggle to identify novel drug targets and to deliver new chemical classes of antibiotics to replace those rendered obsolete by drug resistance. Despite herculean efforts utilizing a wide range of antibiotic discovery platform strategies, including genomics, bioinformatics, systems biology and postgenomic approaches, success has been at best incremental. Obviously, finding new classes of antibiotics is really hard, so repeating the old strategies, while expecting different outcomes, seems to boarder on insanity. The key questions dealt with in this review include: (1) If mutation based drug resistance is the major challenge to any new antibiotic, is it possible to find drug targets and new chemical entities that can escape this outcome; (2) Is the number of novel chemical classes of antibacterials limited by the number of broad spectrum drug targets; and (3) If true, then should we focus efforts on subgroups of pathogens like Gram negative or positive bacteria only, anaerobic bacteria or other group where the range of common essential genes is likely greater?. This review also provides some examples of existing drug targets that appear to escape the specter of mutation based drug resistance, and provides examples of some intermediate spectrum strategies as well as modern molecular and genomic approaches likely to improve the odds of delivering 21st century medicines to combat multidrug resistant pathogens.

## 1. Introduction

Infectious diseases are a leading cause of death worldwide, yet the miracle antibiotics developed to combat them are being lost to the rapid emergence of drug resistance. Modern medicine is heavily dependent on antibiotics that enable many procedures including abdominal surgeries, bone marrow and organ transplants, managing immune compromised patients and cancer patients and for routine hip and knee replacements [1,2]. While one might presume that new antibiotics would be discovered at a sufficient pace to replace those rendered obsolete by resistance, this has not happened. Worse, the few antibiotics that recently entered the clinic are derivatives of older drugs and are susceptible to the same existing drug resistance mechanisms. It is hard to imagine that the last new chemical class of antibiotic for treating infections caused by Gram negative (superbug) bacteria (GNB) was the quinolones, discovered over 50 years ago. And, it is not much better for the Gram positives (GP) with linezolid and daptomycin discovered nearly 30 years ago. While Fleming and Waksman are well known discoverers of antibiotics (penicillin and streptomycin, respectively), it is unlikely that anyone today knows someone that delivered a new class of antibiotic from bench to clinic. Thus, unlike most disciplines of science and technology, there are no experts to advise us. Worse, there seems to be no cumulative knowledge gained from the many failures upon which to build a winning strategy [3].

While it has been argued that the discovery of antibiotics is one of the greatest achievements in medicine in the 20th century; our failure to build a sustainable antibiotic discovery platform has left us with one of the greatest liabilities, and now in the 21st century, one of the greatest challenges. Some blame the lack of funding or of effort, but in retrospect the effort was substantial, particularly by the pharmaceutical industry in the 1990’s, and most agree that massive investments would have made little difference. While much has been written recently to explain these failures [4,5,6,7]; few have provided tactical approaches that might fare any better [6]. Emergence of drug resistance remains the greatest concern and biggest challenge to future discovery efforts. Many believe it is impossible to create an antibiotic to which bacteria cannot evolve resistance and based on some recent experience, the therapeutic window for any new one is projected to be less than five years—a major disincentive. If indeed antibiotic resistance (mutation-based drug resistance) is the major challenge; then, why have we not focused on identifying drug targets less prone to mutation-based resistance? With scarce resources and many suggestions for how to find next generation antibiotics, where do you place your bets [3]? This review examines current discovery approaches and their intrinsic liabilities; challenges some myths associated with antibiotic resistance; and concludes by exploring promising 21st century discovery strategies along with some examples of antibacterials that might escape the specter of drug resistance. There are many headwinds and challenges to antibacterial discovery and delivery of new medicines to the clinic that are not discussed herein (societal, regulatory and marketing) that readers are directed elsewhere for more comprehensive coverage [8,9,10,11].

## 2. Antibacterial Discovery Strategies

Most of the antibiotics used in the clinic today were derived from natural products (NP) identified in screens of environmental material using variations of the Waksman platform strategy [12]. Synthetic antibacterials trace back to Paul Ehrlich who identified chemical compounds like salvarsan that was active in the treatment of Syphilis. Later, Domagk discovered the dye prontosil and its active metabolite sulfanilamide that inhibited folic acid biosynthesis. This work led to development of the sulfa drugs [13]. Medicinal chemistry and the advent of high throughput screens of compound libraries (target based) replaced NP searches in the mid 1990’s and reintroduced the idea of synthetic chemistry as a strategy for creating next generation antibacterials. Advances in computational biology (bioinformatics, systems biology, and metagenomics) and structural biological technologies (crystal structures of protein targets, ligand screens via docking software, the so called designer drug strategy) have led to a third discovery strategy—synthetic biology. While all these technologies engender some optimism, the main reason for the antibiotic discovery void in the first place is that creating new chemical classes is really hard. Is repeating history a viable strategy?—probably not.

## 3. Are Natural Products the Answer?

One definition of insanity is repeating the same task over and over again while expecting a different outcome. This is especially applicable to antibiotic discovery research where much emphasis has been placed on finding new natural product (NP) antibiotics [12]. After all, the vast majority of antibiotics used in the clinic are derivatives of NPs. The insanity with continuing on this path is that for all of these NP antibiotics, resistance mechanisms predated their therapeutic use and this is unlikely to change [3,6,7]. Moreover, we know from past discovery efforts that NPs tend to be redundant—with the same targets and mechanisms of action appearing over and over again [14]. Casting a wider net will not change this reality, because the number of broad spectrum drug targets is quite small [5].

**Hypothesis.** 
*The number of antibiotic classes is limited by the number of broad spectrum drug targets. It follows that the number of resistance mechanisms will generally parallel the number of antibiotic classes. Hence, modifying the genes associated with secondary metabolism, while potentially improving an existing scaffold, is unlikely to produce new chemical classes of antibacterials.*


Another issue with NPs is that since they have evolved in natural environments and not in the human milieu, most NPs are structurally complex and as such are by default poor pharmacophores (PK/PD), requiring much medicinal chemistry to improve bioavailability, to reduce drug metabolism or to reduce intrinsic toxicities. The pharmaceutical industry abandoned NP divisions long ago in favor of high throughput screens of compound libraries against novel drug targets (another failure, see below); or, more productively to create next generation derivatives of existing antibiotics, a strategy that has worked well and continues today as judged by the number of analogues in the current pipeline. Despite the continuum of second, third and fourth generation β-lactams (penicillin, cephalosporin and carbapenem derivatives) [11], by far the largest group of antibiotics used in the clinic today, has not solved the resistance problem originally pointed out by Alexander Fleming in the late 1940’s. While logic dictates that we should abandon further development of this class, advances in pairing them with β-lactamase inhibitors (clavulinate, avibactam, and vaborbactam to mention a few) has restored their effectiveness and extended their clinical lifetimes [15]. Since β-lactamases and β-lactam targets (penicillin binding proteins, PBP) share conserved structural attributes that can be altered via mutation, resistance will eventually defeat them, but for now this strategy is both viable and profitable. The current antibacterial pipeline is populated with new derivatives based on the scaffolds for quinolones, tetracyclines, macrolides and aminoglycosides [16]. New chemistries around existing scaffolds has the potential to overcome current resistance mechanisms by improving binding complexity with the drug target, and or improve pharmacological properties. In this regard, the ketolides were supposed to overcome resistance to macrolides, but the tradeoff is increased toxicities. However, given our experiences with NP antibiotics in general, it is unlikely that continued searches for new ones, regardless of where one looks, will be any more successful than past efforts. 

## 4. Synthetic Antimicrobials—Computational Biology and High Throughput Screens?

In the mid 1990’s, the pharmaceutical industry turned to molecular biology and genomics to sort through microbial genomes for novel drug targets that could be cloned and their products configured into assays amenable to robotic high throughput screens of massive compound libraries (target-based platform). After all, new medicines were found in library screens in other therapeutic areas, including antivirals [17]. The fatal assumption in antibacterial screens was the belief that antibiotic-like material was present in these chemical libraries [3,4,5,6]. In reality, they contained little that resembled antibiotics, because antibiotics are often of high molecular weight and far more complex than can be synthesized by chemists. Worse, most of these compounds were poor pharmacophores (hydrophobic with poor PK/PD properties) and failed both Lipinski’s rule of five and Hergenrother’s rules for penetration of GN bacteria [18,19,20]. Unfortunately, efforts to assemble libraries composed of more “antibiotic like material” have yet to deliver an antibacterial to the clinic. What does “antibiotic like” really mean anyway? The experiences of GlaxoSmithKline, AstraZeneca and others are well documented and today there seems to be little enthusiasm for revisiting this approach [3,5,6]. However, the lists of potential therapeutic drug targets identified in these bioinformatics initiatives and others remain to be exploited. One of the targets in the GSK screen, FabI, an enoyl-ACP reductase and rate-limiting step in fatty acid biosynthesis is the target of a novel therapeutic (Afabicin, formerly Debio-1450) that is active against *Staphylococcus* [21]. Importantly, the failure of HTS to deliver new therapeutics to the clinic should not be confused with whether or not synthetic antibacterials can be developed *de novo* as part of a rational or designer drug strategy (see later section).

## 5. Antibiotic Resistance Theory

Every healthcare worker can tell you why antibiotic resistance is bad for patients and why antimicrobial stewardship (judicious use) is so important, but few have any deep understanding of the diversity of underlying molecular and genetic mechanisms. This has created a “Chicken Little” effect; where strong opinions based on shallow thinking, perpetuation of myths and intolerance of new ideas have unnecessarily complicated the discovery process. Much of antibiotic theory has evolved over nearly 100 years of history and mostly from experiences with NPs. Antibiotics and their respective resistance mechanisms have co-evolved over millions or even billions of years of internecine microbial warfare that is ongoing in the environment. These resistance mechanisms are encoded in DNA and can be disseminated throughout the microbial world via mobile genetic elements and horizontal transfer [22]. It is not surprising that soil and aquatic microbes transfer these resistance determinants to human and animal pathogens, often in clinical settings where antibiotics are in heavy use. There is no good outcome to the continued use of NPs and as discussed previously it is unlikely that new ones will fare any better. This is best exemplified by the emergence of resistance (*mcr-1* family) to colistin, a polymyxin class antibiotic that has been around for decades, but little used, until the emergence of carbapenem resistance in *Klebsiella* (KPC) and other carbapenem resistant enteric superbugs (CRE) made it the drug of last resort [23]. The rapid and global emergence of *mcr-1* just underscores the point. The resistance problem is further exacerbated by globalization, increasing population density and international travel that has brought us *mcr-1* as well as the NDM-1 β-lactamase resistance determinant [24]. The COVID-19 pandemic, unfortunately, is another example of the accelerated global spread of biological agents.

## 6. Synthetic Antimicrobials and Mutation Theory 

It follows from the previous paragraph on NPs, that synthetic antibacterials that inhibit targets “Mother Nature” has yet to find would require millions of years of evolution for resistance determinants to emerge. Note that this hypothesis also holds true for new chemistries against established drug targets, provided that they are also not susceptible to existing resistance mechanisms. A good example is linezolid, a synthetic antibacterial discovered nearly 30 years ago and more recent derivatives (tedizolid), that inhibit protein synthesis in GP bacteria [25]. One might think that synthetic antibacterials would enjoy a long clinical life. Unfortunately, synthetic classes of antibacterials, including linezolid, are often defeated by mutation-based drug resistance. Mutations occur naturally (probability of ~1 in 10^8^) in DNA and since microbial infections involve hundreds of billions of bacteria, microbes tend to win the probability game. Survival is the prime directive for any lifeform and in particular, the stress of antibiotics on bacteria, especially at sub-inhibitory levels, provides strong selection for these resistant variants to emerge and flourish. It should be pointed out that mutation-based drug resistance also underlies resistance for nearly all NPs. This reality led Eric Lander and John P. Holdren to conclude in a summary of their report by the committee on antimicrobial resistance to the President’s Council on Science and Technology that: “In the fight against microbes, no permanent victory is possible: as new treatments are developed, organisms will evolve new ways to become resistant” [26]. This grim prediction is certainly supported by a recent clinical trial of promising synthetic boron containing heterocycle leucyl tRNA synthetase inhibitor which was halted due to rapid emergence of drug resistance [27]. This novel drug inhibited a non-essential proof-reading region of the essential enzyme in which mutations could accumulate and thereby defeat the inhibitor without total loss of function [27]. The lessons learned here include: (1) focus on the essential catalytic center of a prospective drug target, (2) ensure amino acid conservation within this region (resists mutation), and (3) confirm by co-crystallization that leads indeed are binding within the catalytic pocket of the target. It is also helpful early on to explore possible off target activities against human drug targets related to a particular enzyme class. Attention to these details might lead to medicines that slow the inevitable pace to drug resistance. It raises another critical question “can we find drug targets that by catalytic mechanism are capable of escaping mutation-based drug resistance? 

## 7. Targets That Escape Mutation-Based Drug Resistance

Targets that might defeat mutation based drug resistance do exist. One example is the lipid II and lipid III targets associated with cell wall biosynthesis [28]. These targets are composed of lipid and not amino acids and are the target of a new NP antibiotic teixobactin that is in early clinical development for treatment of infections caused by GP bacteria [29]. Another type of target that had not been previously considered is pyruvate: ferredoxin oxidoreductase (PFOR) whose catalytic center is highly conserved through evolution and for which there are no NP inhibitors [30]. The PFOR catalytic mechanism involves a uniquely positioned and contorted vitamin cofactor (vitamin B1 or thiamine pyrophosphate, TPP) [30,31]. Redox cycling via iron/sulfur centers activates the vitamin to enable binding of substrate pyruvate. The nitrothiazolide class of synthetic antibiotics (FDA approved nitazoxanide and analogue amixicile in clinical development) deactivates the vitamin via a proton abstraction mechanism that is dependent on a functional enzyme [30,32]. Two key points here are that mutations to the enzyme (loss of function) or that alter the vitamin, a small molecule, are lethal (2316). TPP is also a cofactor in many enzymes (pyruvate dehydrogenase [PDH] and pyruvate carboxylase), but these are not inhibited by nitazoxanide or amixicile [30,31]. Importantly for nitazoxanide, there are no toxicities in humans or mitochondria which lack the drug target. Importantly, in over 10 years of clinical use, there are no reports of drug resistance with nitazoxanide. A hint at possible resistance to nitazoxanide comes from studies of lab generated resistant mutants of human parasite *Giardia*, where whole genome sequencing found no single correlating mutation among resistant strains [33]. From this study and others, it is possible to suggest that second site compensatory mutations may lead to metabolic shifts that produce a “tolerance like phenotype”, much like over expression of efflux systems incrementally raise MIC for quinolones. Is this evolution at work? It could be, as efflux buys time for mutations in gyrase and topoisomerase genes to manifest. Alternatively, as noted with nitazoxanide tolerance in *Helicobacter pylori*, it may be an example of the microbe “faking it”, i.e., tinkering with limited metabolic regulatory choices to enable survival [31]. In this case, tolerance can be overcome by increasing the therapeutic dose, i.e., one that the microbe cannot overcome by tinkering. The jury is still out though, as only time will tell if such tolerance mechanisms can be inherited. 

## 8. Can we Learn Anything from the Pfor Mechanism?

There are several caveats from the PFOR example that might be useful in identifying new targets: (1) PFOR is an ancient enzyme highly conserved through evolution; (2) the vitamin cofactor is uniquely positioned and tightly bound within the enzyme; (3) the critical amino acids required for cofactor function are absolutely conserved based on analysis of available sequences of PFOR enzymes (>1000) despite whether PFOR is a dimer of high molecular weight monomers or like in *Helicobacter pylori*, a dimer comprised of multiple subunits [34]; and (4) all of these traits are found in the entire broad family of the alpha-keto-acid oxidoreductases (multiple essential targets). Perhaps most important in this example is that nitazoxanide and amixicile work by a “theft” mechanism that requires a functional enzyme and cofactor. The phenotype to the affected pathogen is one of slow starvation (pyruvate) with no definable selective mechanism or means to overcome inevitable death by starvation. 

In retrospect, few searches for new drug targets considered mutation frequency as a criterion for prioritization. Finding catalytic centers that are highly conserved and potentially druggable can be relatively straightforward, but like PFOR, would likely yield targets of limited spectrum. “Highly conserved” generally means that mutations in these regions lead to loss of function or lethality. These potential targets can be catalogued by comparative genomics (BLASTP approach) by comparing genomes (gene products) from a target group of pathogens. By setting the probably of the match (identity) bar to be more stringent (e^−30^ or e^−40^) creates a shorter list of protein matches whose amino acid sequences are highly conserved [35,36]. Further analysis would include amino acid conservation within the catalytic center along with other evaluative criteria that includes chemical space, catalytic mechanism, participation of cofactors and uniqueness from orthologues found in humans or mitochondria. Crystal structure of the enzyme is crucial in this process and in silico screening of chemical libraries becomes an early first step in finding developable leads. A number of recent examples can be used to guide this process [17,37,38]. The power of computational biology and chem-informatics has yet to be fully exploited, so this area has room to grow. An advantage of this approach is that objective criteria replace age old empiric rationale (guesswork) in choosing drug targets. However, the reality of finding new broad spectrum drug targets is close to nil. This is because microbial genomes are small and the major targets for broad spectrum antibiotics are already known (protein synthesis, DNA and RNA synthesis, cell wall biosynthesis and a few central metabolic pathways) [39,40,41,42,43]. As discussed below, we need to consider targets of more limited spectrum. 

## 9. Broad Specrum Versus Narrow Spectrum 

The number of broad spectrum drug targets among microbes that meet the criteria of essential and non-redundant is predicted to be quite small, perhaps fewer than several hundred [5,35]. Most targets are already known and come with the baggage of preexisting antibiotic resistance mechanisms and the reality that they will cause collateral damage to normal flora when deployed. The problem with narrow spectrum antibiotics is that clinicians do not use them when the pathogen responsible for an infection is not known or is potentially polymicrobial. Broad spectrum antibiotics are a hedge and provide liability coverage in the event of an adverse outcome. This reluctance also plagues new antibacterials that enter the clinic, another headwind to drug development. Ironically, getting a new broad spectrum antimicrobial through the Food and Drug Administration (FDA) today is nearly impossible because for each indication a separate clinical trial is generally required and the additive costs are prohibitive. Antibacterials currently in clinical development (gepotidacin) and those that have recently entered the clinic like oritavancin, dalbavancin and tedizolid are examples where the potential use is purposefully limited [5]. In the distant past, the spectrum for an antimicrobial was often established by the process of “off label” use (trial and error), but today, hospital practice/malpractice and liabilities have essentially ended this strategy. We can expect in the future that the number of indications for antibacterials will continue to be limited by development costs [44].

## 10. Limited Spectrum Strategies

Perhaps a more productive antibacterial strategy is to focus on groups of microbes sharing common essential drug targets as exampled by PFOR. In this example, nitazoxanide exhibits the same spectrum as metronidazole, providing broad anaerobic coverage (GP and GN), treatment of infections caused by anaerobic human parasites and potentially treatment of infections caused by members of the epsilon proteobacteria (*Helicobacter* and *Campylobacter*). Along these lines, developing antimicrobials that target either GP or GN bacteria would exploit new drug targets unique to each, such as teixobactin for GPB. Similarly, drugs that are selective against *Mycobacterium tuberculosis* are highly desirable since resistance is a major problem [45]. Recent new chemical classes of anti-tubercular therapeutics include the nitroimadazole delamanid that targets the F_420_ deazaflavin nitroreductase [46]. Similarly, pathogen selective therapeutics such as ridinilazole, a novel synthetic drug in clinical development for treatment of colitis caused by *Clostridioides difficile*, are developed in response to an urgent clinical need [47]. In this case, ridinilazole targets cell division by an unknown mechanism that is likely to uncover a novel drug target and mechanism. Microbiome studies support limited collateral damage with ridinilazole and the frequency of drug resistance appears to be low [48]. 

Since there is an urgent need to develop GN-selective therapeutics, focusing on targets unique to them and vetted by objective criteria (including resistance to mutation) should be encouraged. Indeed, ongoing drug discovery efforts are targeting LPS biosynthesis (LpxC, LepB, MsbA) [49], essential enzymes associated with Beta-Barrel Assembly Machine (BAM), Lipoprotein Assembly (LOL), and even the periplasmic Disulfide Bond Isomerase (DsbA, DsbC, DsbD) enzymes [49,50,51,52,53]. Unfortunately, lead compounds targeting LPS biosynthetic systems have proven toxic and reminds us that finding new chemistries of low toxicity remains a significant challenge. Focus on anti-virulence strategies, particularly for GN superbugs, is now gaining momentum as inhibitors are emerging against secretion systems, pili, motility, quorum sensing and toxins [54,55]. Conceptually, if virulence systems contribute to antibiotic resistance or tolerance, inhibition of these systems might result in regulatory changes that restore or enhance susceptibility to mainline antibiotics. As better rapid diagnostics come onboard, narrow or limited spectrum antibacterials will see increased usage, and when treating polymicrobic infections, therapeutic strategies would likely mix and match the appropriate therapeutics. Even better, we might consider creating bi-functional or hybrid antibacterials that provide dual function that likely will extend both coverage and timelines to resistance. 

Repurposing of drugs, especially FDA approved medicines, for use as antibacterials has gained in interest since they skip the early preclinical and clinical development stages [56]. It is also believed that such drugs would also come with novel modes of action and perhaps escape preexisting drug resistance mechanisms. Perhaps the most studied of these compounds is auranofin, a drug developed to treat rheumatoid arthritis and cancer, that is a potent inhibitor of thioredoxin reductase found in many human pathogens including *Clostridioides difficile* [57]. For repurposed drugs in general the primary indication is still a secondary target of the new application. In order to gain therapeutic efficacy (serum levels greater than MIC), repurposed drugs would likely be administered at concentrations exceeding levels required for efficacy of the original use. In reality, drugs that actively target human enzymes, often come with undesirable side effects or toxicities. Still, for treatment of infections such as tuberculosis, gonorrhea or by *C. difficile* and *H. pylori*, repurposed drugs in combination with other therapeutics might aid overcoming drug resistance. 

## 11. Conclusions 

The goal of this review was to point out basic science challenges to the discovery of new classes of antibacterials and to provide some suggestions on how to move forward in a more productive manner. We know that the “Waksman platform” for discovery of new NPs and the “HTS-target based platform” for screening chemical libraries have both failed to deliver new antibacterials. Does repeating these strategies boarder on insanity? From a business perspective, investment in antibacterial discovery is mitigated by concerns over rapid emergence of drug resistance and loss of investment. If indeed mutation-based drug resistance is the greatest impediment to the discovery effort, then why not work backwards from the problem as a discovery strategy? This review provides a few examples of such drug targets (PFOR and lipid II) and respective inhibitors that shows that this approach has merit. Structure based drug design has yielded therapeutics in other areas including HIV/AIDS (protease inhibitors), antivirals (zanamivir for influenza), and even inhibitors of cyclooxygenase, so why not for antibacterials? Bioinformatics can be used to collate essential targets and provide a probability score for evaluating risk for mutation based drug resistance. Structure based drug design tools have been steadily evolving and their use in evaluating three dimensional structure and chemical space for ligand binding can be integrated with in silico screens of synthetic chemical libraries to produce early leads. Lead optimization and structure activity relationships (SAR) can be modeled and synthetic chemistry and chem-informatics can be used to optimize candidates in areas of PK/PD, drug metabolism and toxicology, essential to building safe medicines. It should be obvious that antibacterial development in the future will require a wide range of biological, chemical, computational and pharmacological disciplines working towards a common goal to progress 21st century medicines. While the pharmaceutical industry used to contain this expertise, we need to find a way to fund such initiatives that also removes the shareholders and corporate return on investment metrics from the equation. While CARB-X and other related initiatives are critical in the clinical development of antibacterials via cost sharing with startup companies, the real bottleneck occurs much earlier in the early discovery process. If 50 years of effort has not yielded a novel class of GN therapeutic, one might imagine the failure rate for any new initiative might be close to 100%, not a viable strategy for “bean” counters. We have to ask “how much are we willing to invest in order to get one new antibiotic to the clinic?” We already know many ways not to make an antibiotic, perhaps now we can find more creative and imaginative ways to produce them. Perhaps if we present the antibiotic crisis as on the scale of the COVID-19 pandemic, maybe it would engender greater attention and resource allocation.

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
