# Peer review of "Antibacterial Discovery: 21st Century Challenges"

_antibiotics, 2020, doi:10.3390/antibiotics9050213_

Round 1

Reviewer 1 Report

Antibacterial Discovery: 21st Century Challenges is a very well written article and by a very senior professor who has a very wide knowledge and provided extensive guidance and suggestions on working in this field if junior members want to enter into this field. 

i request the author to add few lines about drug repurposing. 

Author Response

Response to reviewer’s comments

First, I wish to thank the reviewers for their helpful and constructive comments that have aided me in improving the manuscript.  If you look at how we are collectively responding to this novel coronavirus pandemic, you can see by the development of antivirals and the repurposing of existing medicines as well as the strategies for finding vaccine candidates, this effort parallels what I have tried to call attention to in this review for what is needed to get us the next generation antibacterials to combat drug resistant human pathogens.

Reviewer 1 Antibacterial Discovery: 21st Century Challenges is a very well written article and by a very senior professor who has a very wide knowledge and provided extensive guidance and suggestions on working in this field if junior members want to enter into this field. 

I request the author to add few lines about drug repurposing. 

Response: A section on repurposing drugs was added as requested with an emphasis on auranofin, a widely investigated medicine.

Reviewer 2 Report

The manuscript: "Antibacterial Discovery: 21st Century Challenges" is an interesting, well-written, and needed paper regarding a significant problem, which is the growing resistance of bacteria to ever new groups of antibiotics. It seems necessary to look at the issue as a whole, to indicate both limitations and perspectives in the discovery of new substances with antibiotic activity. However, I have some suggestions to improve the manuscript.

The abstract should be well thought out once. In its current form it is more like a paragraph taken from the introduction section rather than the abstract. The revised abstract should consist rather of the brief summary of the manuscript, namely the main hypothesis, aims, results obtained and the main conclusion.

The work, although very interesting, could be useful to a wider group of recipients if, in the individual paragraphs, in addition to general considerations, there were more examples of specific groups of drugs or single substances (this applies to both natural and synthetic compounds). Without it, the work, although very well written, is general.

Minor editorial errors (line 160 the number for the reference should be provided, the uniformity of in the references section should be maintained).

Based on the comments above I recommend a minor revision of the manuscript.

Author Response

Reviewer 2 The abstract should be well thought out once. In its current form it is more like a paragraph taken from the introduction section rather than the abstract. The revised abstract should consist rather of the brief summary of the manuscript, namely the main hypothesis, aims, results obtained and the main conclusion.

Response: Agree. The abstract was revised as suggested.

The work, although very interesting, could be useful to a wider group of recipients if, in the individual paragraphs, in addition to general considerations, there were more examples of specific groups of drugs or single substances (this applies to both natural and synthetic compounds). Without it, the work, although very well written, is general.

 Response: A few additional examples were added as suggested. The goal of the review though attempted not to get bogged down with too many examples that might detract from the overall emphasis of the review.

Minor editorial errors (line 160 the number for the reference should be provided, the uniformity of in the references section should be maintained).

 Response: Corrected as suggested.

Reviewer 3 Report

The review by Professor Hoffman provides a very accurate and critical analysis of the crisis facing by the discovery process novel antibacteria drugs.
Both techncal and scientific limitations are discussed, but also practical suggestions to overcome these challenges are reported.
The use of grafiphical elements (figures and schemes) and tables is highly recommended.

Author Response

Reviewer 3 The review by Professor Hoffman provides a very accurate and critical analysis of the crisis facing by the discovery process novel antibacteria drugs.
Both techncal and scientific limitations are discussed, but also practical suggestions to overcome these challenges are reported.
The use of grafiphical elements (figures and schemes) and tables is highly recommended.

Response: While in general I agree with the point of adding graphs and tables, I purposefully avoided doing this. From the most recent citations, it is easy to find the examples and what therapeutics are currently in the pipeline. Here, I tried to avoid repeating this information and rather stuck to a focus on a more academic theory behind why we failed in the past and why thinking differently can hopefully get us out of the current mess.